# Comparison of virtual reality development centers and 270-degree evaluations in the context of mid-level managers' competencies

Anna Baczyńska[1]*, Zhenyao Cai[2], Konrad Urbański[3], Łukasz Szajda[3]

**1** Human Resource Department, Kozminski University, Warsaw, Poland, **2** SILC Business School, Shanghai University, Shanghai, China, **3** GuideMe, Warsaw, Poland

* abaczynska@kozminski.edu.pl

## Abstract

Virtual Reality Development Centres (VRDCs) represent a recent extension of Assessment Center (AC) methodology, yet empirical evidence on their validity and their relationship to established evaluation systems remains limited. This study addresses this gap by examining how VRDC assessments—grounded in immersive, real-time behavioral observation—converge with traditional 270-degree evaluations and self-assessments across five managerial competencies. Using a sample of 64 mid-level managers who completed 16 VRDC sessions and parallel 270-degree evaluations, we tested three hypotheses concerning convergent validity and unique diagnostic value. VRDC demonstrated strong inter-rater reliability (rwg = .82–.95; ICC(2) =.76–.92) and showed significant alignment with 270-degree ratings for managing people and tasks, goal orientation, and change management, but not for decision-making or cooperation. Self-assessments correlated with VRDC only for cooperation, and negatively for managing people and tasks, revealing consistent self-perception biases. Across all competencies, VRDC provided diagnostic insights not captured by self-report, supporting its added theoretical value. The findings contribute to theory by clarifying the distinct construct domains captured by immersive behavioral simulations versus retrospective, perception-based evaluations. We argue that VRDC should be conceptualized not merely as a technological enhancement, but as a methodological bridge that integrates AC logic with multi-source frameworks. Practically, VRDC offers organizations a reliable and context-sensitive tool for assessing crisis-relevant competencies, complementing—but not replacing—traditional evaluation methods. The study advances the theoretical understanding of VR-based assessment and informs the development of multimethod competency assessment systems.

**Data availability statement:** The dataset is publicly available in Zenodo at DOI: 10.5281/zenodo.18187535 (https://doi.org/10.5281/zenodo.18187535).

**Funding:** The author(s) received no specific funding for this work.

**Competing interests:** The authors have declared that no competing interests exist.

## Introduction

In contemporary organizations, leaders are expected to navigate environments that are not only volatile and uncertain but also structurally complex and institutionally fragmented. This so-called VUCA context fundamentally reshapes what it means to be competent at work, shifting the focus from static skills to adaptive and context-sensitive capabilities that enable individuals and teams to respond effectively to rapid change [1]. Competency assessment has therefore become a central pillar of human resource development, linking individual career trajectories with organizational performance, resilience, and long-term employability [2]. Over recent decades, methodological advances have led to the widespread adoption of multi-source feedback systems such as 180-, 270-, and 360-degree evaluations, as well as sophisticated behavioral methods such as Assessment Centers (ACs), which are widely regarded as "gold-standard" tools for assessing managerial potential in high-stakes contexts [3; 4,5].

At the same time, accelerating digitalization and the emergence of advanced simulation technologies have opened new possibilities for enhancing the ecological validity, standardization, and analytical depth of competency assessment [6]. Virtual reality (VR) offers highly immersive and interactive environments in which complex social, cognitive, and emotional demands can be modeled with remarkable fidelity while preserving experimental control [7; 8]. Against this backdrop, Virtual Reality Development Centres (VRDCs) have emerged as a promising innovation that integrates the core logic of the Assessment Center methodology with the technological affordances of VR. VRDCs enable the simulation of dynamic, ambiguous, and high-pressure managerial situations that are difficult—or impossible—to recreate within traditional in-person or digitally mediated AC formats. In doing so, they expand the methodological toolkit available to researchers and HR practitioners who seek to diagnose and develop managerial competencies in a manner that is both rigorous and contextually relevant.

This article positions VRDC within the broader tradition of behavioral assessment methods, with particular emphasis on Assessment Centers and multi-source feedback systems. It argues that VRDC can serve as a methodological and conceptual bridge between behavioral simulations and perception-based evaluations, offering a multi-method, multi-source framework for triangulating managerial competencies. By comparing VRDC outcomes with 270-degree evaluations and self-assessments, the present study contributes to both the theoretical understanding and practical application of immersive simulations in competency assessment. It also explores the extent to which VRDC can complement, and in some cases partially replace, traditional assessment systems, thereby contributing to the ongoing modernization of human development practices in contemporary organizations.

### Assessment centers

The Assessment Center (AC) has long been recognized as one of the most valid and comprehensive methods for evaluating managerial competencies and identifying

developmental potential. Its strength lies in the direct observation of behavior across multiple simulation-based exercises that reflect the complexity of real organizational challenges [3; 4,9,10]. Unlike self-reports or perception-based measurement tools, ACs allow for a multidimensional, behaviorally anchored evaluation of participants' decisions, problem-solving strategies, and interpersonal dynamics under standardized yet realistic conditions. The theoretical foundation of ACs is rooted in the principle that behavior exhibited in structured simulations generalizes to behavior in real work environments, provided that exercises are designed to elicit the target competencies [11,12].

Traditionally, ACs serve both diagnostic and developmental purposes. From a diagnostic standpoint, they provide validated assessments of competencies. From a developmental standpoint, they identify areas for growth, guide individual development plans, and serve as a structured basis for feedback dialogues with participants [13]. ACs typically include tasks such as leaderless group discussions, in-basket exercises, role plays, strategic analyses, and crisis simulations, each designed to provide independent evidence of competency manifestations. Trained assessors evaluate participants using structured behavioral indicators aligned with competency models, enhancing both reliability and construct validity.

When AC outcomes are combined with multi-source feedback, particularly 360-degree feedback, practitioners gain deeper insight into the alignment—or misalignment—between observed behaviors in standardized simulations and long-term behavioral patterns perceived in the workplace [14,15]. The AC provides a "snapshot" of behavior in controlled conditions, while 360-degree feedback reflects relational dynamics and sustained behavior patterns. Research has shown that correlations between AC and 360-degree results tend to be modest, underscoring that each method captures unique facets of performance [16]. For developmental purposes, this complementarity is invaluable.

Development Centers (DCs), which stem from the same methodological roots as ACs, emphasize developmental feedback even more directly. Their reliability indices—often reflected in intraclass correlation coefficients ranging from 0.81 to 0.92—demonstrate strong inter-rater agreement when the methodology is implemented rigorously [17,18]. However, despite their strengths, traditional ACs face limitations related to cost, scalability, logistics, and standardization. High-quality ACs require multiple assessors, carefully designed exercises, physical space, and extensive coordination, which makes them difficult to scale globally. These limitations have accelerated the shift toward digital, hybrid, and VR-based assessment environments.

## Virtual Reality Development Centres (VRDC)

Virtual Reality Development Centres (VRDCs) represent a significant advancement in assessment methodology, extending the AC logic into immersive, technologically enhanced environments. A VRDC can be conceptualized as a virtual behavioral assessment platform that replicates realistic work scenarios within fully immersive 3D spaces. Participants, equipped with VR headsets, enter simulated managerial contexts requiring strategic thinking, prioritization, interpersonal coordination, crisis decision-making, and complex problem resolution. Their behaviors are recorded, analyzed, and evaluated by trained assessors based on predefined competency models, following psychometric standards aligned with AC/DC methodology [19].

The use of VR introduces several distinctive advantages over both traditional in-person and digital ACs. First, VRDCs offer heightened realism and immersion, eliciting spontaneous behaviors that reflect the emotional and cognitive complexity of real work situations. Participants operate within environments that simulate spatial, temporal, and interpersonal dynamics more faithfully than two-dimensional digital simulations. This increased ecological validity enhances the authenticity of behavioral data collected during the assessment [7].

Second, VRDCs enable a high degree of standardization, ensuring that all participants face identical conditions regardless of location or time. This level of environmental control minimizes contextual variability that can influence behavior in traditional assessments, strengthening fairness and comparability across individuals and cohorts.

Third, VR makes possible the precise capture of micro-behaviors—such as gaze direction, reaction time, gesture patterns, tone of voice, interpersonal distance, or physiological cues—that are difficult to capture reliably in live simulations

[8]. These data streams can be recorded unobtrusively and subjected to advanced analytics, including machine learning models, enabling more granular insights into performance patterns and behavioral dynamics.

Fourth, VRDCs allow for the safe simulation of high-stakes or ethically sensitive scenarios, such as conflict escalation, system failures, or emergency response. Simulating such scenarios in the physical world may introduce safety, ethical, or operational risks; VRDCs eliminate these constraints while preserving realism.

Fifth, VRDCs support scalable talent assessment systems. Once developed, VR simulations can be deployed widely across geographies at relatively low marginal cost, making VRDCs particularly attractive for global organizations seeking consistent, scalable assessment processes.

Nevertheless, VRDC implementation also poses significant challenges. Development involves high initial costs for hardware, software, scenario design, and expert collaboration among psychologists, HR specialists, software engineers, and learning designers. Some participants may experience motion sickness or discomfort in immersive environments [20]. Psychometric validation is essential to ensure that VRDC outcomes demonstrate reliability and construct validity comparable to traditional ACs. Furthermore, VRDC scenarios may not capture long-term behavioral patterns or sustained team dynamics as effectively as workplace-based evaluations, requiring careful methodological integration with other assessment systems.

## Multi-source evaluation methods

Multi-source feedback systems, including 180-degree, 270-degree, and 360-degree assessments, are fundamental tools in competency assessment and leadership development. They gather evaluative input from multiple stakeholders, capturing diverse perspectives on an employee's strengths, weaknesses, and developmental needs. The 180-degree evaluation relies primarily on self-assessment and direct supervisor feedback. It is efficient and simple to implement but offers a limited perspective on interpersonal and relational competencies [21]. The 270-degree evaluation adds input from peers or direct reports, providing a more complex social perceptual lens. Research indicates that this approach supports nuanced self-awareness and deepens developmental conversations [22,23]. The 360-degree evaluation, the most comprehensive option, includes feedback from supervisors, colleagues, subordinates, and sometimes clients. This holistic perspective makes it particularly valuable for assessing leadership, relational intelligence, collaboration, and influence [24].

However, multi-source systems share limitations. Their accuracy depends on organizational culture, willingness to provide honest feedback, and evaluator training. They are subject to halo effects, leniency biases, political influences, and other perceptual distortions [25]. Importantly, multi-source assessments are inherently retrospective, capturing impressions accumulated over time rather than real-time behavior in dynamic or high-pressure contexts. As such, they cannot fully evaluate competencies such as crisis decision-making, adaptability, or rapid prioritization. This limitation creates fertile ground for integrating multi-source feedback with behavioral assessment methods such as ACs and VRDCs.

## Integration: Linking VRDC, assessment centers, and multi-source feedback

Research on the relationship between multi-source feedback and AC results shows that although the two methods capture complementary dimensions of performance, correlations between them tend to be modest [26; 27]. ACs capture behavior under controlled conditions, while multi-source evaluations capture perceptions of behavior over time and across contexts. The combination of both provides a more holistic and valid assessment of managerial competence.

VRDCs represent a methodological bridge that unites these traditions. By simulating realistic, dynamic, and emotionally engaging work situations, VRDCs generate behavioral indicators that are simultaneously contextually grounded and standardized. Immersive technologies appear particularly promising for reducing certain forms of bias, enhancing ecological validity, and increasing participant engagement [28]. When VRDC results are triangulated with 270-degree feedback

and self-assessments, they create a multi-layered competency model that includes (1) objective, behavior-based observations collected in real time; (2) social and relational insights from the workplace; and (3) introspective perspectives on self-awareness, perceived competence, and readiness.

Discrepancies between these sources—such as overestimation in self-perception, under-recognition of strengths by peers, or inconsistencies between situational and habitual behavior—offer valuable insights into self-awareness, behavioral consistency, and development potential [29]. Such triangulation aligns with contemporary human development perspectives advocating multi-method assessment to enhance construct validity and developmental impact [30; 31; 32].

### Research gaps in competency assessment using virtual reality

Despite the growing interest in immersive technologies within the field of human resource development (HRD), the scholarly literature still lacks sufficiently coherent and empirically grounded evidence on the role of Virtual Reality Development Centres (VRDCs) in the assessment of managerial competencies. As outlined earlier in the introduction, VRDCs represent a promising evolution of traditional Assessment Centers by combining their behavioral foundations with the high ecological validity, standardization, and measurement precision afforded by immersive virtual environments [7; 8]. However, despite these theoretical advantages, a number of critical research gaps remain, limiting the full integration of VRDCs into mainstream competency assessment methodologies.

A first and significant gap concerns the lack of comparative studies juxtaposing VRDCs with well-established assessment methods such as 270-degree evaluations or traditional Assessment Centers. While ACs are widely recognized for their strong construct validity [33,12] and multi-source evaluations capture long-term perceptions of workplace behavior [24], very few studies systematically examine whether VRDCs produce convergent, divergent, or complementary competency patterns relative to these methods. Importantly, the broader AC literature has long been criticized for insufficient methodological transparency: many published AC studies do not report crucial design and procedural variables that influence the reliability and validity of the method [34,35,36]. This lack of detailed reporting makes it even more difficult to determine how VRDCs compare to ACs, and creates substantial challenges for cumulative research, including meta-analytic synthesis.

A second research gap relates to the psychometric properties of VRDC assessments. Traditional ACs consistently demonstrate high reliability, with intraclass correlation coefficients often ranging from.81 to.92 [18]. By contrast, little is known about whether VRDCs meet comparable standards of reliability, construct validity, and criterion-related validity. Although VR environments yield rich streams of behavioral data—such as gaze patterns, reaction times, nonverbal cues, and spatial behaviors—immersion and realism alone do not guarantee measurement validity [6,37]. Rigorous validation efforts are therefore needed before VRDCs can be considered equivalent to or superior to classical behavioral assessment methodologies.

A third gap concerns the limited understanding of which managerial competencies are most effectively assessed through VR. While scholars frequently suggest that VR may be particularly well-suited for diagnosing dynamic competencies such as crisis decision-making, conflict management, adaptability, or prioritization, empirical studies rarely specify the mechanisms through which VR affords more accurate assessment of these competencies relative to ACs or perception-based evaluations [38]. Without systematic evidence, it remains unclear which competencies VRDCs genuinely assess more effectively—and whether these competencies are reliably predictive of workplace performance.

A fourth gap arises from the absence of longitudinal research evaluating the durability and predictive validity of VR-based assessments. Whereas classical ACs and 360-degree evaluations possess well-documented prognostic validity [14,15], studies on VRDCs are almost exclusively cross-sectional. It remains unknown whether behaviors exhibited in immersive simulations translate into real-world professional behaviors or whether VRDCs serve as valid predictors of future performance, leadership potential, or developmental progression.

A fifth gap pertains to participants' experiences with VR and their potential influence on performance outcomes. Unlike traditional ACs, VR introduces technological and physiological factors—such as motion sickness, comfort with immersive

equipment, emotional reactivity to realistic scenarios, or cognitive overload—that may influence both behavior and perceived fairness of the assessment process [39]. Understanding these experiential variables is essential for interpreting VRDC outcomes and designing simulations that minimize measurement distortion.

A sixth gap revolves around the lack of integration of VRDCs within multimodal assessment systems, despite widespread agreement in the HRD literature that multimethod triangulation—combining behavioral observation, social perception, and self-reflection—produces the most valid and developmentally meaningful competency assessments [40; 32]. Most existing VR research treats VRDCs as stand-alone tools rather than components of an integrated, multi-source system. This isolates VRDCs from established organizational assessment practices and prevents understanding of how VR-generated data align with or diverge from traditional evaluation results.

In light of these gaps, there is a clear need to examine VRDC outcomes in conjunction with self-assessments and multi-source feedback, assessing discrepancies and alignments across behavioral (VRDC), perceptual (270-degree), and introspective (self-report) data. Such an approach makes it possible to identify the unique diagnostic value of VRDCs and supports the development of modern, multimodal competency assessment systems.

### Research questions and hypotheses

Given the research gaps identified above, the present study focuses on analyzing the relationships among VRDC results, 270-degree evaluations, and self-assessments. Self-assessments, while easy to administer and widely used, are subject to well-known distortions resulting from limited self-awareness and self-enhancement tendencies [41,24]. VRDCs, by contrast, rely on direct observation of behavior in immersive, high-fidelity simulations that replicate demanding managerial contexts, thereby revealing performance gaps that are not evident in declarative self-ratings [7; 38].

Building on this logic, the following research questions guide the present study:

**RQ 1. Is there a correlation between VRDC assessments and the ratings obtained from the 270-degree evaluation?**

**RQ 2. Which of the evaluations—others' assessments (managers/employees) or self-appraisals—positively or negatively correlate with VRDC assessments?**

To address these questions, the following hypotheses are proposed:

**H1. There is a significant correlation between VRDC assessments and the ratings obtained from the 270-degree evaluation.**

**H2. Both others' assessments and self-appraisals show a positive correlation with VRDC assessments.**

**H3. VRDC assessments provide insights into competencies that are not captured by self-assessments.**

By triangulating VRDC results with 270-degree evaluations and self-assessments, this study empirically examines the extent to which VRDCs offer diagnostic value that is distinct from, or complementary to, perception-based methods, directly addressing several of the research gaps identified above and contributing to the advancement of contemporary competency assessment practices.

## Materials and methods

### Participants and procedure

The study was approved by the Research Ethics Committee of Kozminski University, which issued written approval for its conduct. All participants were fully informed about the purpose of the study and subsequently provided their informed written consent to participate. All subjects participated voluntarily. The Declaration of Helsinki was adequately addressed.

A total of 64 participants, aged between 25 and 51 years (mean age = 33.30, SD = 4.56), with 54% being male, took part in this study. The company where the study was conducted was responsible for recruiting participants and distributing written informed consent forms for participation. The recruitment period for this study took place from 01/03/2024–30/03/2024.

Data on managerial Dimensional Competencies (DC) were collected using the Virtual Reality Development Center (VRDC) method. We organized 16 VRDC sessions for managers. Our VRDC adhered to the guidelines outlined in the International Taskforce on Assessment Center Guidelines. [19].

The performance in the Assessment Center (AC) was determined based on the aggregated consensus ratings of assessors for each AC dimension. Such consensus, often referred to as 'staff meeting' judgments, is a standard criterion for evaluating AC performance, as suggested by [3]. All assessors were qualified psychologists. Subsequently, both the participants and their evaluators completed a 270-degree evaluation questionnaire online.

## Measures

**Virtual reality development center.** The "Breakdown in the Power Plant" scenario serves as an advanced diagnostic tool designed to assess and develop key professional competencies in a simulated, high-stakes environment. By integrating the methodology of the Assessment Center (AC) into a virtual reality (VR) setting, this tool offers a dynamic and immersive approach to competency evaluation. Participants are introduced to a routine meeting with a power plant manager, only to face an unexpected and critical breakdown upon arrival. This sudden crisis sets the stage for a series of five interconnected exercises, each aimed at evaluating specific competencies under realistic and controlled conditions.

The scenario emphasizes five primary competencies critical to effective professional functioning: Managing people and tasks, goal orientation, decision-making, change management, and cooperation.

1. **Managing People and Tasks:** Participants are tasked with managing teams under crisis conditions, focusing on role assignment, action coordination, and maintaining team cohesion. Leadership is tested through scenarios that require participants to remain focused and make decisions under significant time pressure to achieve objectives efficiently.

2. **Goal Orientation:** Clearly defined tasks demand strategic thinking, prioritization, and resource allocation. This competency assesses participants' determination and ability to consistently work towards goals, even when faced with technical and temporal constraints.

3. **Decision-Making:** Scenarios such as selecting city districts for power disconnection or managing cooling water resources evaluate participants' ability to analyze situations rapidly, foresee consequences, and employ strategic solutions based on limited or incomplete data.

4. **Change Management:** Dynamic and evolving conditions, such as sudden reactor pressure surges or new information displayed on control screens, require participants to demonstrate flexibility and adaptability. Their responses to real-time challenges are critically evaluated.

5. **Cooperation and social skills:** Participants engage in tasks requiring effective communication, active listening, information sharing, and mutual support. The ability to build consensus and collaborate effectively in high-pressure situations is a central focus.

The competency model used in this study was developed on the basis of a systematic job analysis of the mid-level manager position in the organization and calibrated against the organization's competency framework, in line with international assessment center guidelines.

**Scenario.** The simulation unfolds in a structured yet adaptive manner, beginning with what seems to be a routine professional interaction. However, an unforeseen breakdown quickly escalates into a high-stakes crisis, compelling participants to act decisively.

The scenario incorporates five stages:

1. **Accessing the Control Room:** Participants must assemble an access key using fragments of codes found on badges, testing their organizational skills and ability to coordinate as a team.

2. **Managing Critical Technical Parameters:** Once inside the control room, participants address key operational challenges, such as adjusting reactor cooling levels, managing power output, and controlling pressure within cooling towers.

3. **Cooling System Management:** This stage requires strategic handling of water resources across three tanks, each with unique constraints. Participants must balance immediate needs with long-term consequences, emphasizing strategic thinking.

4. **Power Output Decisions:** Participants analyze a map displaying city districts and must decide which areas to disconnect from the power supply to reduce reactor load. This task combines technical considerations with social implications, highlighting ethical decision-making.

5. **Crisis Resolution:** In the final stage, participants respond to a critical reactor pressure surge. They must quickly evaluate available options and either stabilize the system or initiate a shutdown. This stage demands cohesive teamwork and rapid decision-making under pressure.

The simulation leverages VR technology to deliver immersive and realistic crisis scenarios. Information critical to decision-making is conveyed through innovative tools such as virtual watches and control panel screens, faithfully replicating modern crisis management systems. The dynamic nature of the tasks allows the simulation to adapt to participants' actions, enhancing realism and providing deeper insights into their competencies.

This tool was developed to diagnose and enhance participants' competencies in managing real-world challenges. By combining immersive VR technology with rigorous evaluation criteria, the scenario offers a comprehensive assessment of participants' abilities to perform under pressure, adapt to changing environments, and collaborate effectively.

Example of our VRDC we showed in the links below:

1. https://youtu.be/Tj3DxC7n8kA

2. https://youtu.be/9ppIKCTa2Fk

3. https://youtu.be/a0QycvZkRxU

We assessed five competencies: Managing people and tasks, Goal orientation, Decision Making, Change management, Cooperation and Social Skills.

We used scale, where:

• **Level A – desired behaviors were not observed,** undesirable behaviors or behaviors that do not bring the desired results dominate; there is consistency of undesirable behavior in different situational contexts.

• **Level B -desired behaviors dominated;** consistent behavior desired **in standard situations** was observed; in difficult or complex situations, the person could exhibit ineffective behavior or achieve incomplete results.

• **Level C – the best patterns dominated in all situation**.

We have divided our scale into smaller intervals: A=0; B-=1, B=2, B+=3, C-=4, C=5.

The Virtual Reality Development Centre (VRDC) followed standardized assessment procedures in accordance with the International Taskforce on Assessment Center Guidelines [19] and was implemented by two trained assessors. The assessor training lasted two days and included both theoretical and practical components. The theoretical module introduced the competency framework, rating procedures, and behavioral indicators, while the practical module involved calibration exercises using pilot VRDC sessions to ensure inter-rater consistency.

Assessors evaluated participants' behaviors using a Behaviourally Anchored Rating Scale (BARS) The Assessment Center used the BARS, in which assessors did not rate competencies directly during the exercise but instead recorded

detailed behavioral incidents observed throughout each task. Only after the full Assessment Center session were these behavioral records grouped, interpreted, and rated under the appropriate competency categories. This sequencing reflects a long-standing recommendation within the Assessment Center literature that emphasizes separating behavioral observation from competency judgment to improve accuracy and reduce context-specific bias [42,33,12]. By relying on raw behavioral data accumulated across multiple exercises, BARS supports higher construct validity and stronger inter-rater reliability, as competencies are inferred from broader and more representative sets of behavioral evidence. Recent conceptual analyses of AC methodology further reinforce the value of this approach, noting that delayed judgment based on aggregated behavioral indicators yields clearer construct measurement and more stable ratings across assessors [43].

Each competency was operationalized through observable behavioral indicators describing performance at six incremental levels. The behavioral rating scale included the categories A = 0, B− = 1, B = 2, B+ = 3, C− = 4, and C = 5. Each category incorporated behavioral anchors to guide assessor judgment, and assessors were instructed to select the level that most accurately reflected the participant's demonstrated performance rather than assigning intermediate values. All five competencies were assessed consistently across all five stages of the VRDC scenario.

**270-degree evaluation.** We created a questionnaire, identical to the one employed in the Virtual Reality Development Method, to assess the same five competencies. This questionnaire featured the same 64 items and was based on a 5-point scale.(see attachment 1).

The 270-degree assessment employed the same competency framework and rating scale as the VRDC but was based on workplace performance rather than simulated behavior. Each participant was evaluated by one manager and two co-workers. Prior to completing the questionnaire, all raters received written instructions explaining the definitions of each competency, the rating scale anchors, and examples of relevant workplace behaviors. Raters were asked to base their judgments on observed behaviors over the previous six months and to complete the assessments independently and confidentially.

The use of a unified model ensured conceptual consistency between the VRDC and 270-degree assessments, enabling meaningful comparison of results across both methods.

## Data analyses

Data were analyzed in SPSS Statistics version 26 and Microsoft Excel. First, $r_{wg}$ and ICC were calculated to check the inter-rater reliability of the ratings obtained from self assessment and VRDC assessments. Second, bivariate correlations were run to check associations between the study variables to test the proposed hypotheses. All responses were coded, no identifiable information was stored, and access to raw data was restricted to the research team in compliance with institutional ethics guidelines.

## Results

### Preliminary analyses

As employee assessments made by others were collected from three individuals: 1 manager, 2 employees/co-workers, it was necessary to justify whether the three others' assessments of the same employee tend to be stable and consistent. To do this, we conducted a $r_{wg}$ test to assess the level of inter-rater agreement for the assessments from three others. Rwg is a measure of within-group agreement, indicating the extent to which ratings from different assessors converge. A value above 0.70 is considered acceptable.

The samples with low $r_{wg}$ were excluded and a final sample of 52 employees, who were middle level of managers, was included in further analysis. Similarly, since we also collected employee assessments from two Assessors from VRDC, the $r_{wg}$ of VRDC assessments were calculated. All $r_{wg}$ of VRDC assessments were above the cutoff of 0.70, indicating the inter-rater agreement of VRDC assessments was desirable [18].

In addition, an analysis of variance (ANOVA) was performed to examine between-group variations and compute the intraclass correlation coefficient (ICC) of others assessments and VRDC, respectively, to reflect the inter-rater reliability [17]. ICC (Intraclass correlation coefficient) represents the level of agreement between different raters. Higher ICC values indicate greater reliability and consistency in assessments.

For the 270-degree evaluations (Table 1), $r_{wg}$ values ranged from 0.84 to 0.91, all exceeding the recommended threshold of 0.70 [18], indicating acceptable within-group agreement across all competencies. ICC(1) values ranged from 0.17 to 0.64, and ICC(2) values ranged from 0.38 to 0.84. While most competencies demonstrated adequate reliability, Decision Making showed lower ICC values (ICC(1) = 0.17, ICC(2) = 0.38), suggesting less consistency in rater assessments for this competency. These all ICC(1) and ICC(2) values were within the range of desirable norms [17,18]. Sample sizes varied across competencies (N = 45–51) due to the exclusion of participants with low inter-rater agreement ($r_{wg} < 0.70$).

For VRDC assessments (Table 2), all reliability indices were within desirable ranges. The $r_{wg}$ values ranged from 0.82 to 0.95, ICC(1) values from 0.62 to 0.85, and ICC(2) values from 0.76 to 0.92, confirming strong inter-rater reliability across all competencies. All 64 participants were included in the VRDC analyses.

To visualize these patterns across all competencies, Fig 1 presents a comprehensive comparison of correlations and reliability indices across the five assessment methods. The figure illustrates both the convergence and divergence between VRDC, self-assessments, and others' assessments, while also highlighting the reliability characteristics of each method.

As shown in Fig 1A, the pattern of correlations varied substantially across competencies, with others' assessments showing stronger alignment with VRDC than self-assessments for most competencies. The reliability indices (Fig 1B

**Table 1. Inter-rater Reliability Indices for 270-Degree 'Others' Assessments.**

| Competency | rwg | ICC(1) | ICC(2) | N |
|---|---|---|---|---|
| Managing People and Tasks | 0.89 | 0.64 | 0.84 | 51 |
| Goal Orientation | 0.86 | 0.47 | 0.73 | 47 |
| Change Management | 0.91 | 0.49 | 0.75 | 49 |
| Decision Making | 0.84 | 0.17 | 0.38 | 45 |
| Cooperation and Social Skills | 0.90 | 0.45 | 0.71 | 50 |

*Notes.* $r_{wg}$ = within-group agreement index; ICC(1) = intraclass correlation coefficient representing proportion of variance due to participant differences; ICC(2) = reliability of mean ratings across raters; N = number of participants after excluding those with low inter-rater agreement ($r_{wg} < 0.70$). Participants with missing data were retained in the sample but not included in reliability calculations.

**Table 2. Inter-rater Reliability Indices for VRDC Assessments.**

| Competency | rwg | ICC(1) | ICC(2) | N |
|---|---|---|---|---|
| Managing People and Tasks | 0.89 | 0.82 | 0.90 | 64 |
| Goal Orientation | 0.82 | 0.62 | 0.76 | 64 |
| Change Management | 0.86 | 0.68 | 0.81 | 64 |
| Decision Making | 0.87 | 0.78 | 0.88 | 64 |
| Cooperation and Social Skills | 0.95 | 0.85 | 0.92 | 64 |

*Notes.* $r_{wg}$ = within-group agreement index; ICC(1) = intraclass correlation coefficient representing proportion of variance due to participant differences; ICC(2) = reliability of mean ratings across raters; N = total number of participants.

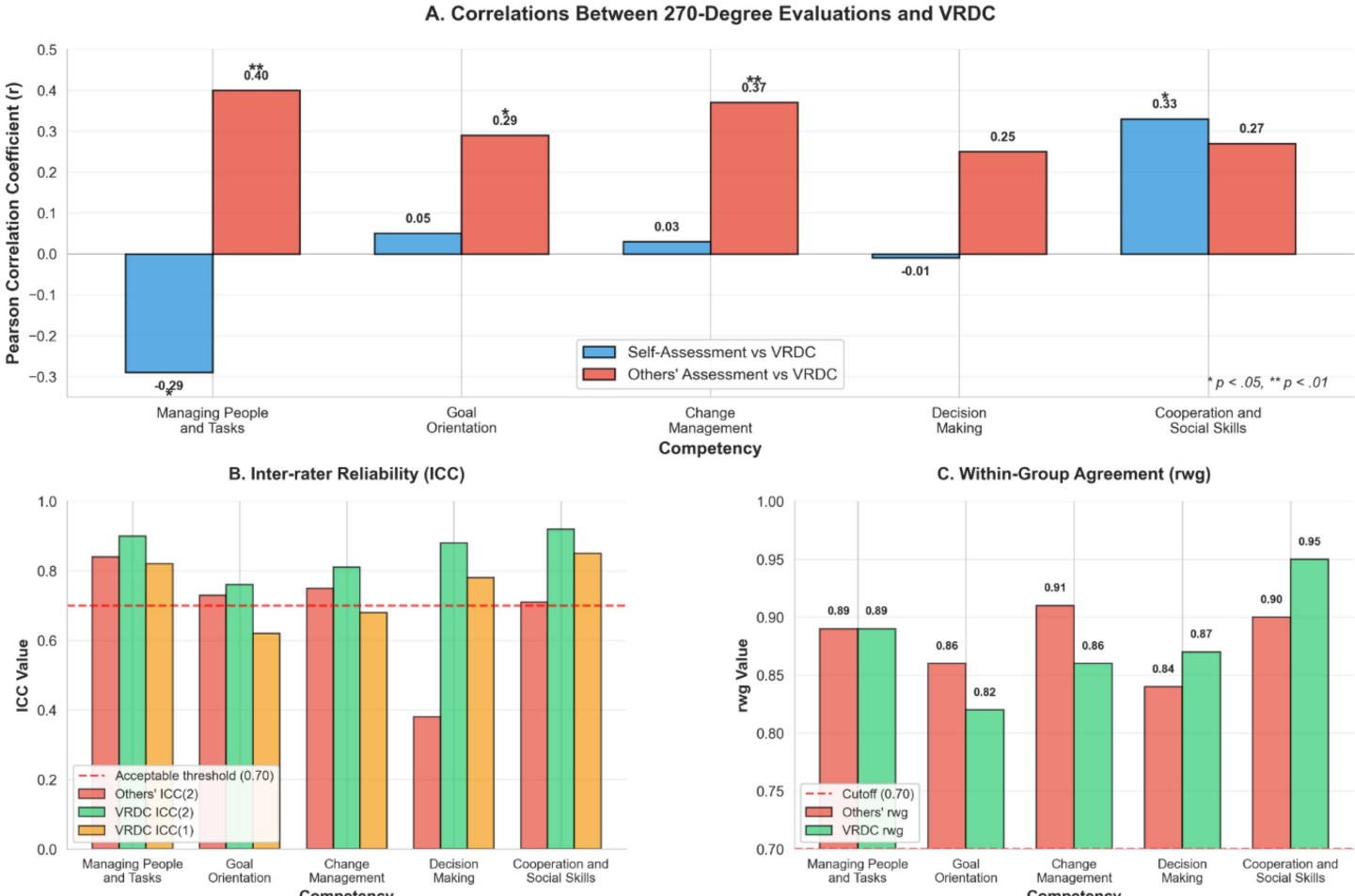

**Fig 1. Comprehensive comparison of assessment methods showing correlations and reliability indices across five managerial competencies.** Panel A displays Pearson correlation coefficients between self-assessments (blue bars) and others' assessments (red bars) with VRDC performance. Significant positive correlations between others' assessments and VRDC were observed for Managing People and Tasks (r = 0.40, p < .01), Goal Orientation (r = 0.29, p < .05), and Change Management (r = 0.37, p < .01). Self-assessments showed negative correlation with VRDC for Managing People and Tasks (r = −0.29, p < .05) and positive correlation for Cooperation and Social Skills (r = 0.33, p < .05). Panel B presents inter-rater reliability using intraclass correlation coefficients, showing that VRDC assessments consistently exceeded the acceptable threshold of 0.70 for ICC(2) across all competencies, while others' assessments showed lower reliability for Decision Making (ICC(1) = 0.17, ICC(2) = 0.38). Panel C illustrates within-group agreement ($r_{wg}$) values, with all competencies exceeding the 0.70 cutoff for both assessment methods, indicating acceptable rater agreement. * p < .05, ** p < .01.

and 1C) demonstrate that VRDC maintained consistently high inter-rater reliability across all competencies, whereas 270-degree assessments showed more variability, particularly for Decision Making.

## Correlation analyses

To test the research hypotheses, the relationships between the study variables were examined using bivariate correlations. Pearson product-moment correlation coefficients were calculated to examine the relationships between study variables (Fig 1A). Table 3 presents the interrelationship among employee self-assessments, the ratings obtained from others,

**Table 3. Correlations coefficients between self-assessments, others assessments, and VRDC assessments for Managing people and tasks (M).**

|  | 1 | 2 | 3 |
|---|---|---|---|
| (1) Self-assessments | – |  |  |
| (2) Others assessments | 0.19 | – |  |
| (3) VRDC assessments | −0.29* | 0.40** | – |
| M | 3.78 | 3.68 | 3.32 |
| SD | 0.58 | 0.84 | 1.03 |

*Notes.* *$p < .05$, **$p < .01$ two-tailed. All items were measured on a 5−point Likert scale.

and VRDC assessments in M. Regarding the evaluation of M, employee self-assessments were significantly and negatively correlated with VRDC assessments ($r = −0.29$, $p < 0.05$), and others assessments were significantly and positively correlated with VRDC assessments ($r = 0.40$, $p < 0.01$). Therefore, H1 and H2 were supported. These results suggest that participants tend to underestimate their own capabilities in managing people and tasks compared to their performance observed in the VRDC simulations. In contrast, ratings from managers and colleagues align more closely with VRDC assessments, indicating that others' perceptions of participants' managerial competencies are more congruent with their actual demonstrated behaviors in high-stakes, realistic scenarios.

Table 4 shows the interrelationship among employee self-assessments, the ratings obtained from others, and VRDC assessments in G. For the evaluation of G, others assessments and VRDC assessments were positively correlated ($r = 0.29$, $p < 0.05$), supporting H1. However, employee self-assessments were not significantly correlated with VRDC assessments, indicating H2 was not supported. The positive correlation between others' assessments and VRDC suggests that managers and colleagues can accurately evaluate participants' goal-directed behaviors based on workplace observations. However, the lack of correlation between self-assessments and VRDC indicates that participants may have limited self-awareness regarding their actual goal orientation in demanding situations, or that self-perceptions of goal-setting abilities do not necessarily translate into effective goal-directed performance under pressure.

Table 5 shows the interrelationship among employee self-assessments, the ratings obtained from others, and VRDC assessments in CH. In terms of CH, others assessments were positively related with VRDC assessments ($r = 0.37$, $p < 0.01$), supporting H1. In addition, employee self-assessments were not significantly correlated with VRDC assessments. Therefore, H2 was not supported. The significant positive correlation between others' ratings and VRDC assessments demonstrates that change management capabilities, as perceived by managers and colleagues in everyday work contexts, correspond well with adaptive behaviors demonstrated in simulated crisis scenarios. The absence of correlation with self-assessments suggests that individuals may find it challenging to accurately evaluate their own flexibility and adaptability to change, particularly in high-pressure situations that deviate from routine experiences.

**Table 4. Correlations coefficients between self-assessments, others assessments, and VRDC assessments for Goal orientation (G).**

|  | 1 | 2 | 3 |
|---|---|---|---|
| (1) Self-assessments | – |  |  |
| (2) Others assessments | 0.25 | – |  |
| (3) VRDC assessments | 0.05 | 0.29* | – |
| M | 3.90 | 3.60 | 3.81 |
| SD | 0.50 | 0.76 | 0.89 |

*Notes.* *$p < .05$, **$p < .01$ two-tailed. All items were measured on a 5−point Likert scale.

**Table 5. Correlations coefficients between self-assessments, others assessments, and VRDC assessments for Change management (CH).**

|  | 1 | 2 | 3 |
|---|---|---|---|
| (1) Self-assessments | – |  |  |
| (2) Others assessments | 0.34* | – |  |
| (3) VRDC assessments | 0.03 | 0.37** | – |
| M | 3.88 | 3.64 | 3.83 |
| SD | 0.57 | 0.74 | 0.89 |

*Notes.* $^*p < .05$, $^{**}p < .01$ two-tailed. All items were measured on a 5−point Likert scale.

Table 6 demonstrates the interrelationship among employee self-assessments, the ratings obtained from others, and VRDC assessments in D. Results revealed that both employee assessments and self-appraisals made by others were not significantly correlated with VRDC assessments. Therefore, H1 and H1 were not supported. The lack of significant correlations for decision-making competency indicates that VRDC captures unique aspects of decision-making under time pressure and uncertainty that are not reflected in traditional 270-degree evaluations or self-assessments. This suggests that decision-making in simulated crisis scenarios (where immediate consequences unfold in real-time) may involve different cognitive processes or behavioral patterns than decision-making in routine work settings. The low inter-rater reliability for this competency in 270-degree assessments (ICC(1) = 0.17, ICC(2) = 0.38, see Table 1) may also contribute to the lack of convergence with VRDC ratings.

Table 7 presents the interrelationship among employee self-assessments, the ratings obtained from others, and VRDC assessments in C. As results indicated, employee self-assessments were positively correlated with VRDC assessments (r = 0.33, p < 0.05) for the evaluation of C, while others assessments were not related with VRDC assessments. Therefore,

**Table 6. Correlations coefficients between self-assessments, others assessments, and VRDC assessments for Decision Making (D).**

|  | 1 | 2 | 3 |
|---|---|---|---|
| (1) Self-assessments | – |  |  |
| (2) Others assessments | 0.35* | – |  |
| (3) VRDC assessments | −0.01 | 0.25 | – |
| M | 3.99 | 3.58 | 3.77 |
| SD | 0.57 | 0.76 | 1.08 |

*Notes.* $^*p < .05$, $^{**}p < .01$ two-tailed. All items were measured on a 5−point Likert scale.

**Table 7. Correlations coefficients between self-assessments, manager assessments, and VRDC assessments for Cooperations (C).**

|  | 1 | 2 | 3 |
|---|---|---|---|
| (1) Self-assessments | – |  |  |
| (2) Others assessments | 0.22 | – |  |
| (3) VRDC assessments | 0.33* | 0.27 | – |
| M | 4.14 | 3.87 | 4.19 |
| SD | 0.45 | 0.53 | 0.84 |

*Notes.* $^*p < .05$, $^{**}p < .01$ two-tailed. All items were measured on a 5−point Likert scale.

H1 and H1 were not supported. Interestingly, for cooperation and social skills, self-assessments showed a positive correlation with VRDC performance, while others' assessments did not. This pattern suggests that participants may have accurate self-awareness of their collaborative tendencies and social competencies, which manifest consistently across both self-reports and observed behaviors in team-based VRDC scenarios. The lack of correlation with others' assessments may reflect that cooperative behaviors in short-term, high-intensity VRDC simulations differ from long-term collaborative patterns observed by colleagues in everyday work contexts.

According to the above correlation analysis results, Table 8 summarizes the results of hypothesis testing.

## Discussion

The present study examined the extent to which Virtual Reality Development Centres (VRDCs) converge with, diverge from, and complement traditional 270-degree evaluations and self-assessments across five managerial competencies. By juxtaposing immersive, real-time behavioral observations with retrospective social evaluations and introspective judgments, the study extends prior research on the relationships between simulation-based methods and multi-source feedback [33,25]. The findings show that VRDC is neither a technological replica of established tools nor an outright substitute; rather, it provides a distinct yet partially overlapping perspective on managerial competence.

### VRDC and 270-degree evaluations: Competency-specific convergence

Regarding RQ1 and H1, the results indicate partial convergence between VRDC and 270-degree evaluations. Significant positive correlations emerged for managing people and tasks, goal orientation, and change management, suggesting that performance in immersive simulations reflects behavioral tendencies visible to managers and co-workers in the workplace. This pattern aligns with well-established evidence that behavioral simulations and multi-source ratings tend to show modest but meaningful overlap [27; 44].

For decision-making and cooperation, however, convergence was absent. This is consistent with two plausible interpretations. First, competencies such as crisis decision-making may manifest differently in acute high-pressure VR scenarios than in routine, distributed workplace decisions captured by 270-degree evaluations. Second, the low ICC values for decision-making in others' ratings suggest that the absence of correlation may be partly attributable to inconsistency among raters rather than a lack of validity in VRDC. These findings reinforce earlier critiques that multi-source evaluations are vulnerable to perceptual biases, relational histories, and political dynamics [25], whereas VRDC captures immediate behavioral responses unfiltered by social interpretation.

**Table 8. Results of hypothesis test.**

| Dimension | H1 | H2 | H3 |
|---|---|---|---|
| M | Supported | Supported | Supported |
| G | Supported | Not supported | Supported |
| CH | Supported | Not supported | Supported |
| D | Not supported | Not supported | Supported |
| C | Not supported | Not supported | Supported |

*M – managing people and task.

G – goal orientation.

D – decision making process.

CH – change management.

C – Cooperation.

Overall, convergent validity appears to be competency-dependent. VRDC aligns with 270-degree evaluations when competencies are stable, frequently observed, and strongly embedded in daily work processes, but diverges when competencies are context-sensitive or shaped by social dynamics.

### Alignment with self- and others' assessments: Asymmetrical patterns

Findings related to RQ2 and H2 point to a marked asymmetry: VRDC aligns more strongly with others' ratings than with self-assessments, except in the domain of cooperation.

The negative correlation between VRDC and self-ratings for managing people and tasks highlights well-documented tendencies toward self-enhancement, particularly in domains central to professional identity [24]. Lack of alignment between VRDC and self-evaluations in goal orientation and change management suggests that self-perceptions in dynamic contexts are often inaccurate or overly abstract, failing to predict real-time behavioral adaptation.

Cooperation uniquely showed a positive VRDC–self link with no convergence between VRDC and others' ratings. This indicates that participants may possess relatively accurate introspective insight into their immediate collaborative behavior in simulated tasks, while colleagues' judgments reflect longer-term relational patterns, political dynamics, or organizational norms. The result nuances commonly held assumptions about the universal inaccuracy of self-assessments and supports emerging findings that self-insight may vary substantially across competency types.

### Unique diagnostic value of VRDC

Hypothesis 3 was supported across all competencies. VRDC identified strengths and weaknesses not reflected in self-assessments, demonstrating its ability to reveal behavioral blind spots. Because VRDC captures spontaneous reactions to realistic, pressure-laden events, it provides access to cognitive–emotional patterns that retrospective methods cannot easily detect. This supports theoretical claims that immersive simulations enhance ecological validity and reveal behavior that is otherwise difficult to observe [7; 38].

The consistently high rwg and ICC values for VRDC further strengthen the methodological case for VR-based assessment, suggesting that the immersive format does not compromise psychometric rigor. By contrast, the variability of 270-degree reliability—especially for decision-making—highlights perennial challenges in perception-based ratings and underscores the value of supplementing them with behavioral data.

### VRDC as a complementary rather than substitutive method

Taken together, the results suggest that VRDC is best conceptualized as a complementary method within multimodal assessment systems. For competencies such as managing people and tasks, goal orientation, and change management, VRDC can serve as a partial substitute for multi-source feedback when organizational constraints limit the feasibility of 270-degree processes. However, for competencies requiring complex, context-dependent, or relationally embedded behaviors—especially decision-making under uncertainty and long-term cooperation—VRDC captures dimensions that 270-degree evaluations overlook.

This aligns with broader arguments in the literature emphasizing the value of multi-method, multi-source triangulation to capture the full behavioral, perceptual, and introspective spectrum of managerial competence [40; 32]. VRDC enriches this ecosystem by providing standardized, high-fidelity behavioral evidence that complements both subjective perceptions and self-reports.

### Methodological considerations

While the findings are promising, they should be interpreted in light of several limitations. The sample was modest in size and restricted to mid-level managers in a single organizational context, limiting generalizability. VRDC results reflect

performance in a single crisis-simulation scenario, which may accentuate certain behavioral patterns at the expense of others. The uneven psychometric quality of the 270-degree instrument complicates interpretation of non-convergence. Finally, the cross-sectional design precludes conclusions about predictive validity—a crucial dimension for establishing the long-term value of VR-based assessment.

**Future research directions should incorporate:**

1. Larger and cross-cultural samples to evaluate generalizability.

2. Multiple VR scenarios to broaden construct coverage.

3. Longitudinal designs examining predictive validity for leadership effectiveness, promotion, and learning outcomes.

4. Multimodal data integration, including physiological and behavioral analytics from VR (e.g., gaze, reaction time).

5. Individual difference moderators, such as cognitive style, VR experience, and emotional regulation.

## Conclusion

Overall, the study demonstrates that VRDC is a psychometrically robust, behaviorally sensitive, and diagnostically valuable tool for assessing managerial competencies. VRDC contributes unique insights beyond those obtained from 270-degree evaluations and self-assessments, particularly in high-stakes decision-making and dynamic collaboration. Rather than replacing established methods, VRDC enriches competency assessment by adding a realistic, standardized, and cognitively demanding layer of behavioral evidence, making it a powerful component of contemporary talent management systems.

## Supporting information

**S1 Data. DATA_description.**
(DOCX)

**S1 File. Questionnaire_270 degree assessment.**
(DOCX)

## Author contributions

**Conceptualization:** Anna Baczyńska, Zhenyao Cai, Łukasz Szajda, Konrad Urbański.

**Data curation:** Anna Baczyńska, Łukasz Szajda, Konrad Urbański.

**Formal analysis:** Anna Baczyńska, Konrad Urbański.

**Methodology:** Anna Baczyńska, Zhenyao Cai, Konrad Urbański.

**Supervision:** Zhenyao Cai.

**Validation:** Łukasz Szajda.

**Writing – original draft:** Anna Baczyńska, Zhenyao Cai, Łukasz Szajda, Konrad Urbański.

**Writing – review & editing:** Zhenyao Cai, Łukasz Szajda, Konrad Urbański.

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
