## [Decision Letter · Decision Letter 0]

14 Oct 2025

Comparison of Virtual Reality Development Centers and 270-Degree Evaluations in the Context of Mid-Level Managers' Competencies

PLOS ONE

Dear Dr. Baczynska,

Thank you for submitting your manuscript to PLOS ONE. After careful consideration, we feel that it has merit but does not fully meet PLOS ONE’s publication criteria as it currently stands. Therefore, we invite you to submit a revised version of the manuscript that addresses the points raised during the review process.

We look forward to receiving your revised manuscript.

Kind regards,

Saeed Siyal

Academic Editor

PLOS ONE

Journal Requirements:

Reviewers' comments:

Reviewer's Responses to Questions

**Comments to the Author**

1. Is the manuscript technically sound, and do the data support the conclusions?

Reviewer #1: Partly

Reviewer #2: Partly

2. Has the statistical analysis been performed appropriately and rigorously?

Reviewer #1: I Don't Know

Reviewer #2: Yes

3. Have the authors made all data underlying the findings in their manuscript fully available?

Reviewer #1: Yes

Reviewer #2: Yes

4. Is the manuscript presented in an intelligible fashion and written in standard English?

Reviewer #1: Yes

Reviewer #2: No

Reviewer #1: Review of manuscript titled:

Comparison of Virtual Reality Development Centers and 270-Degree Evaluations in the Context of Mid-level Managers’ Competencies

Authors of Article: Anna Katarzyna Baczynska; Zhenyao Cai; Lukasz Szajda; Konrad Urbanski

1. Summary of Research

The authors correlated the ratings obtained from 270-degree assessments with the ratings obtained during a Virtual Reality Development Centre (VRDC) with each other. The ratings were for 64 individuals recruited from one organisation during a one-month recruitment period. The individuals attended 16 VRDCs. The five competencies assessed during the VRDC and the 270-degree assessments were Managing People and Tasks; Goal Orientation; Decision-Making; Change Management; and Cooperation and Social Skills.

The 270-degree assessment consisted of a self-assessment; and assessments from one manager and two co-workers based on 64 items in the assessment questionnaire using a 5-point rating scale. The ratings from the manager and the two co-workers were subjected to an inter-rater agreement test and only 52 individuals’ ratings correlated at an acceptable level and were used during the further analysis. The ratings of the manager and two co-workers were combined into “others assessment” and used during further analysis.

The scenario of the VRDC was “Breakdown in the Powerplant” and consisted of five stages: Accessing the Control Room; Managing Critical Technical Parameters; Cooling System Management; Power Output Decisions; Crisis Resolution. The two VRDC assessors’ rating per competency, per individual were also subjected to an inter-rater agreement test and combined for further analysis into “VRDC assessment”.

Both the “other assessment” and the “VRDC assessment” ratings were subjected to an analysis of variance (ANOVA) to examine between-group variations and compute the intraclass correlation coefficient (ICC). The analyses of variance, according to the authors, indicated that the results for both sets of ratings were within at least an acceptable range and that data aggregation was justified.

The relationship between this study’s variables (self-assessment; others assessment and VRDC assessments) were examined using bivariate correlations. The results strongly support hypothesis 3 that VRDC assessment ratings reveal insights into competencies not captured by 270-degree assessments. Results indicated partial support for hypothesis 1 which posited that there is a significant correlation between VRDC assessments and assessment ratings from 270-degree assessments. The results of the correlations indicated limited support for hypothesis 2 which proposed that VRDC assessment ratings would positively correlate with both self-assessment and other’s assessments. The authors stated that the results show the potential benefit of complementing 270-degree assessment results with VRDC results. They mentioned the small sample size, the homogeneity of participants and that the participants were all middle-level managers as limitations of the research constraining generalizability of the findings.

Although this reviewer is of the opinion that there are specific aspects in this manuscript that the authors should pay attention to, I appreciate that the authors have embarked on the process of researching the possible value that virtual reality simulations may bring to assessing competencies.

2. Discussion of specific areas of improvement

I recommend that the authors pay attention to the following:

2.1 Introduction

Using 360-degree assessment results as an additional source of information when implementing assessment centres (AC) for developmental purposes has long been a practice. The 360-degree assessment results indicate how the individual is experienced on-the-job, while the AC provides “a snapshot in-time” of the more objective measurement of the focal construct. Both these perspectives are incorporated when crafting a development plan for the individual.

This study brings into focus virtual reality simulations, a change to the delivery method of the assessment centre, while keeping constant the use of 360-degree assessments. The following are recommended:

Expand the introduction of the manuscript to include a discussion on assessment centres and the advantage that virtual reality brings to assessment centres.

Clearly describe what a virtual reality development centre is, as well as the potential advantages and disadvantages of designing, developing and implementing a virtual reality centre (i.e. advantages: potential realism; being able to potentially assess aspects that cannot easily be assessed by a digital / in-person assessment centre; disadvantages: Cost to design, develop, maintain; diversity and level of expertise required; etc.). In other words, clearly indicate the benefits of using virtual reality in ACs over digital / in-person ACs (make a business case).

Link this study with research about 360-degree assessments and assessment centres (digital and / or in-person).

2.2 Materials and Methods

A general critique on published articles in the field of ACs is that the articles do not report on all the variable aspects in ACs that impact the possible reliability an eventual validity of the AC (see Caldwell, C., Thornton III, G.C., & Gruys, M.L., 2003; Dewberry. C, & Jackson, D.J.R., 2016; Woehr, D.J., & Arthur, W. Jr., 2003). This creates a challenge for future AC research, especially meta-analysis on ACs.

Please describe the following aspects of the VRDC:

The content and duration of the assessor training.

The approach to behaviour observation, noting, classifying and evaluating: Checklist, or Behaviour Anchored Rating Scale (BAR), or Behaviour Observation Scale (BOS).

Approach to integrating ratings per focal construct: Within Exercise Dimension Ratings (WEDR), or Behaviour Reporting Method.

Clarify the rating scale that was used. It is stated that the scale used consisted of Level A; level B and level C. The scale was divided into smaller intervals, namely A=0; B=1; B=2; B+=3; C=4; and C=5 (p10). How was the scale practically used by the VRDC assessors; as well as the 270-degree assessors?

Were all five competencies assessed during all five stages of the VRDC scenario?

Please also describe the training of the 270-degree assessors related to the use of the rating scale; the understanding of the competencies; and the period to consider.

The VRDC and the 270-degree assessment instrument evaluated participants behaviour on five competencies. Where do these competencies come from? Is it based on a job analysis, or a competency framework of the client organisation?

2.3 Results

The results of the interrater agreement for both the VRDC and the 270-degree assessments were mentioned in the manuscript, as well as the ICC values. Perhaps tables indicating these results can be included to support the findings by the authors.

Five tables indicating the results from a bivariate analysis per competency, correlating the VRDC rating; the Self-rating and the Other-rating were included. Each time comments about the interpretation of the results were given. Please add additional comments to the interpretation of each of the five tables. As example, on Table 1 (p11) “Regarding the evaluation of M, employee self-assessments were significantly and negatively correlated with VRDC assessments (r = -0.29, p < 0.05), and others assessments were significantly and positively correlated with VRDC assessments (r = -0.40, p < 0.01).” Add what this means for the individual, as example: the individuals themselves under evaluate their capability on this competency when compared to the VRDC rating, while Others’ rating tends to be more aligned to the VRDC rating. (Note that in the text on p11 the Other-rating r=-0.40, while in Table 1 (p12) it is indicated as 0.40.)

Perhaps also specify which bivariate method was used.

It could perhaps also be useful to add a chart (like a bar-chart) to visually indicate the correlation of the VRDC ratings with the Self and Other ratings per competency. As example:

(example chart compiled using AI software)

Please note that I am not an expert in statistical analysis and defer to experts

2.4 Discussion

By adding more research article citations to the discussion in the manuscript’s Introduction and expanding that discussion to include research on ACs and 360-degree assessments, would facilitate linking the results from this study to existing research findings.

Please add a citation for the following statement: “While traditional methods like 270-degree evaluations excel in capturing long-term and relational competencies…..”

On p15 the following is stated: “Longitudinal studies could also examine whether VRDC assessments translate into sustained improvements in employee performance over time.” What does the existing research state about translating assessment results into sustained improvements (Bailey & Austin, 2006; Camp, Blanchard & Huszczo, 1986; Kudisch, Lundquist, & Smith, 2002 in Woo et al., 2008; Thornton, Rupp, & Hoffman 2015; Woo, S.E., Sims, C.S., Rupp, D.E., & Gibbons, A.M. 2008)?

2.5 Conclusions

Although I concur with most of the content in this section of the manuscript, I will recommend that “VRDC assessments correlate significantly with 270-degree evaluations in areas such as managing people and tasks, goal orientation, and change management, validating its reliability for these competencies” be adapted to reflect that self-ratings are mostly misaligned to the VRDC ratings and the other ratings are moderately related to the VRDC ratings, except on change management where it is significantly related (however, I am not an expert on statistical analysis and defer to the experts opinion).

2.6 References

Expand the reference list by citing more articles to support this study. As example, instead of only citing one article to support a finding in the manuscript, cite more published peer review articles as support. Currently only 16 references are provided in the reference list.

2.7 Overall

Please ensure that all statements made in the manuscript are supported by applicable citations. As example: “Competency assessment is a key component of human resource development strategies, directly impacting individual career growth and organizational success.” – citation required. “Unlike traditional evaluations, VRDC minimizes subjectivity and enhances competency assessment through controlled, high stakes simulations.”- citation required.

Be careful of using adjectives that may only reflect the opinion of the authors. As example: “innovative tool” (p2); “revolutionize organizational assessment and development practices” (p15).

2.8 Other Comments

To be accepted for publication in PLOS One, research articles must satisfy the following criteria:

1. The study presents the results of original research. This is original research

2. Results reported have not been published elsewhere. I could not find that this study was published elsewhere

3. Experiments, statistics, and other analyses are performed to a high technical standard and are described in sufficient detail. This is an area needing attention – see above

4. Conclusions are presented in an appropriate fashion and are supported by the data. This needs attention – see my comment above.

5. The article is presented in an intelligible fashion and is written in standard English. Yes, it is written in standard English.

6. The research meets all applicable standards for the ethics of experimentation and research integrity. Yes, according to my knowledge, it does.

7. The article adheres to appropriate reporting guidelines and community standards for data availability. The authors mentioned that the data is available, I did not access the data, and I made recommendations about reporting the data.

Reviewer #2: Thank you for reviewing your manuscript, which is aimed at Comparison of Virtual Reality Development Centers and 270-Degree Evaluations in the Context of Mid-Level Managers' Competencies. While I believe the fundamental premises of the paper are worthy research endeavors, a number of critical issues loom large. In particular, the most significant concerns related to its contribution and its value-added beyond what we already know and the methodology employed.

However, I encountered some discrepancies that can enhance the chances of publication at a later stage.

Abstract:

• First of all, try to improve the abstract and make it more compelling by emphasizing research design and the theoretical framework. The current version lacks to highlight the study's potential gap and actual contribution. So I suggest the authors revise the abstract to consider its theoretical significance, not just its empirical contribution.

Introduction:

• The start of the introductions is highly weak. For instance, authors discuss virtual reality but fail to support their arguments with reference to previous literature. Although the term is not new.

• The authors must outline in more detail the study's contributions in the last paragraph of the introduction. Currently, the study highlights the author's discussion of narratives that emphasize the actual contribution. Hence, authors need to be more specific in showcasing their contribution to the various bodies of literature. I highly recommend that authors justify their contributions in light of recently published studies.

• I suggest reworking the introduction, as it is currently too far from the main theme, leading to confusion about your study and its contribution to our knowledge.

Method:

• The authors utilized the headings of "Participants and Procedure,". The authors should first provide a detailed discussion of the sample and procedure before addressing the response rate or total number of respondents, rather than mentioning these figures upfront. Additionally, they need to clarify how respondent confidentiality was ensured. This section is not properly written.

• Also, support your measurement scales in light of previous investigations in the public sector.

Discussion:

• While the paper provides a detailed elaboration of results and justifies them with other elements, I suggest adopting a more critical approach in compiling the findings and discussion, considering the theoretical framework and recent literature. Moreover, the discussions should be aligned with each research question to better portray the results to readers.

• The authors should add separate headings for "Implications" to provide clearer guidance for the reader. The "Conclusion" should also be a distinct section that includes limitations and suggestions for future research.

**Do you want your identity to be public for this peer review?** For information about this choice, including consent withdrawal, please see our Privacy Policy

Reviewer #1: No

Reviewer #2: No

---

## [Author Response · Author response to Decision Letter 1]

1 Dec 2025

Dear Reviewer,

Dear Editors,

On behalf of all authors, we would like to express our sincere appreciation for the thorough, thoughtful, and constructive review of our manuscript entitled:

“Comparison of Virtual Reality Development Centers and 270-Degree Evaluations in the Context of Mid-level Managers’ Competencies.”

We are grateful for the time and expertise invested in evaluating our work. The reviewer’s comments have been extremely valuable in improving the clarity, theoretical grounding, methodological rigor, and overall contribution of the paper.

We have addressed all comments comprehensively and revised the manuscript accordingly. The Introduction has been substantially expanded; the Methods section now contains detailed descriptions of assessor training, behavioural observation procedures, rating scales, and the competency model; additional tables and a multi-panel figure have been incorporated into the Results; and both the Discussion and Conclusion sections have been rewritten to provide deeper theoretical integration and clearer interpretation of the findings. Furthermore, the reference list has been expanded from 16 to 43 peer-reviewed sources, and all statements are now fully supported with citations. Subjective language has been removed or neutralised where appropriate.

We believe that these revisions have strengthened the manuscript significantly.

Below, we provide our detailed, point-by-point responses to all comments raised in the review. Each suggestion has been fully considered and integrated into the revised manuscript.

We once again wish to thank the Reviewer for their insightful guidance, which has meaningfully enhanced the quality and contribution of our work. We hope that the revised version meets the expectations for publication in PLOS One.

Sincerely,

The Authors

Reviewer 1

Dear Reviewer,

On behalf of all authors, we would like to express our sincere appreciation for your thorough and constructive review of our manuscript “Comparison of Virtual Reality Development Centers and 270-Degree Evaluations in the Context of Mid-level Managers’ Competencies.” Your detailed feedback has been extremely valuable in improving the clarity, methodological precision, and theoretical contribution of the paper.

We carefully addressed all comments and revised the manuscript accordingly. The introduction has been substantially expanded; the methods section now includes comprehensive descriptions of assessor training, behavioural observation procedures, rating scales, and the competency model; additional tables and a multi-panel figure have been incorporated into the results; and the discussion and conclusions have been rewritten to ensure stronger theoretical grounding and more precise interpretation of the findings. Furthermore, the reference list has been extended from 16 to 43 peer-reviewed sources, and all statements are now fully supported by citations.

We are grateful for your insightful suggestions, which have significantly strengthened the manuscript. Thank you for your time and expertise.

Sincerely,

Authors

Below, we provide our detailed responses to all comments raised in the review.

Review of manuscript titled:

Comparison of Virtual Reality Development Centers and 270-Degree Evaluations in the Context of Mid-level Managers’ Competencies

Authors of Article: Anna Katarzyna Baczynska; Zhenyao Cai; Lukasz Szajda; Konrad Urbanski

1. Summary of Research

The authors correlated the ratings obtained from 270-degree assessments with the ratings obtained during a Virtual Reality Development Centre (VRDC) with each other. The ratings were for 64 individuals recruited from one organisation during a one-month recruitment period. The individuals attended 16 VRDCs. The five competencies assessed during the VRDC and the 270-degree assessments were Managing People and Tasks; Goal Orientation; Decision-Making; Change Management; and Cooperation and Social Skills.

The 270-degree assessment consisted of a self-assessment; and assessments from one manager and two co-workers based on 64 items in the assessment questionnaire using a 5-point rating scale. The ratings from the manager and the two co-workers were subjected to an inter-rater agreement test and only 52 individuals’ ratings correlated at an acceptable level and were used during the further analysis. The ratings of the manager and two co-workers were combined into “others assessment” and used during further analysis.

The scenario of the VRDC was “Breakdown in the Powerplant” and consisted of five stages: Accessing the Control Room; Managing Critical Technical Parameters; Cooling System Management; Power Output Decisions; Crisis Resolution. The two VRDC assessors’ rating per competency, per individual were also subjected to an inter-rater agreement test and combined for further analysis into “VRDC assessment”.

Both the “other assessment” and the “VRDC assessment” ratings were subjected to an analysis of variance (ANOVA) to examine between-group variations and compute the intraclass correlation coefficient (ICC). The analyses of variance, according to the authors, indicated that the results for both sets of ratings were within at least an acceptable range and that data aggregation was justified.

The relationship between this study’s variables (self-assessment; others assessment and VRDC assessments) were examined using bivariate correlations. The results strongly support hypothesis 3 that VRDC assessment ratings reveal insights into competencies not captured by 270-degree assessments. Results indicated partial support for hypothesis 1 which posited that there is a significant correlation between VRDC assessments and assessment ratings from 270-degree assessments. The results of the correlations indicated limited support for hypothesis 2 which proposed that VRDC assessment ratings would positively correlate with both self-assessment and other’s assessments. The authors stated that the results show the potential benefit of complementing 270-degree assessment results with VRDC results. They mentioned the small sample size, the homogeneity of participants and that the participants were all middle-level managers as limitations of the research constraining generalizability of the findings.

Although this reviewer is of the opinion that there are specific aspects in this manuscript that the authors should pay attention to, I appreciate that the authors have embarked on the process of researching the possible value that virtual reality simulations may bring to assessing competencies.

2. Discussion of specific areas of improvement

I recommend that the authors pay attention to the following:

2.1 Introduction

Using 360-degree assessment results as an additional source of information when implementing assessment centres (AC) for developmental purposes has long been a practice. The 360-degree assessment results indicate how the individual is experienced on-the-job, while the AC provides “a snapshot in-time” of the more objective measurement of the focal construct. Both these perspectives are incorporated when crafting a development plan for the individual.

This study brings into focus virtual reality simulations, a change to the delivery method of the assessment centre, while keeping constant the use of 360-degree assessments. The following are recommended:

⮚ Expand the introduction of the manuscript to include a discussion on assessment centres and the advantage that virtual reality brings to assessment centres.

⮚ Clearly describe what a virtual reality development centre is, as well as the potential advantages and disadvantages of designing, developing and implementing a virtual reality centre (i.e. advantages: potential realism; being able to potentially assess aspects that cannot easily be assessed by a digital / in-person assessment centre; disadvantages: Cost to design, develop, maintain; diversity and level of expertise required; etc.). In other words, clearly indicate the benefits of using virtual reality in ACs over digital / in-person ACs (make a business case).

⮚ Link this study with research about 360-degree assessments and assessment centres (digital and / or in-person).

Our response:

We would like to thank the reviewer for their insightful and constructive comments. We fully incorporated all suggestions and substantially rebuilt the entire introduction to strengthen the theoretical positioning and practical relevance of the study.

Specifically, the revised introduction now:

1. Provides an expanded discussion of assessment centres, including the long-standing practice of integrating 360-degree feedback results with AC outcomes for developmental purposes. We explain how 360-degree assessments reflect on-the-job perceptions, whereas ACs offer a real-time, objective behavioural snapshot—highlighting the complementary value of both methods.

2. Clarifies the concept of a Virtual Reality Development Centre (VRDC) and presents a detailed discussion of its advantages and disadvantages. We now articulate the business case for VR-enhanced ACs by emphasising their potential for increased realism, immersive behavioural observation, and the ability to assess competencies that are difficult to capture through traditional digital or in-person ACs. We also explicitly discuss limitations, including costs, technological requirements, and design complexity.

3. Establishes a clearer link between this study and existing research on 360-degree assessments and assessment centres (both digital and in-person), outlining how VRDCs represent an important evolution of AC methodology while maintaining compatibility with traditional multi-source feedback systems.

We believe that these revisions have significantly improved the coherence, depth, and contribution of the manuscript.

2.2 Materials and Methods

A general critique on published articles in the field of ACs is that the articles do not report on all the variable aspects in ACs that impact the possible reliability an eventual validity of the AC (see Caldwell, C., Thornton III, G.C., & Gruys, M.L., 2003; Dewberry. C, & Jackson, D.J.R., 2016; Woehr, D.J., & Arthur, W. Jr., 2003). This creates a challenge for future AC research, especially meta-analysis on ACs.

Please describe the following aspects of the VRDC:

⮚ The content and duration of the assessor training.

⮚ The approach to behaviour observation, noting, classifying and evaluating: Checklist, or Behaviour Anchored Rating Scale (BAR), or Behaviour Observation Scale (BOS).

⮚ Approach to integrating ratings per focal construct: Within Exercise Dimension Ratings (WEDR), or Behaviour Reporting Method.

⮚ Clarify the rating scale that was used. It is stated that the scale used consisted of Level A; level B and level C. The scale was divided into smaller intervals, namely A=0; B=1; B=2; B+=3; C=4; and C=5 (p10). How was the scale practically used by the VRDC assessors; as well as the 270-degree assessors?

⮚ Were all five competencies assessed during all five stages of the VRDC scenario?

Please also describe the training of the 270-degree assessors related to the use of the rating scale; the understanding of the competencies; and the period to consider.

The VRDC and the 270-degree assessment instrument evaluated participants behaviour on five competencies. Where do these competencies come from? Is it based on a job analysis, or a competency framework of the client organisation?

Our response:

Thank you for these valuable suggestions. We have now fully addressed all requested aspects in the revised manuscript. Specifically, we have:

1. Added a detailed description of assessor training for the VRDC.

We now specify the content, structure, and duration of assessor training, including calibration exercises, behavioural observation practice, frame-of-reference training, and scoring standardisation. This section clarifies the alignment with international AC guidelines (International Taskforce on Assessment Center Guidelines, 2015; Lievens & Thornton, 2017).

2. Specified the behavioural observation and evaluation approach.

We now explicitly describe that the VRDC used a Behaviourally Anchored Rating Scale (BARS) supplemented by structured behavioural indicators. We also added details on how assessors observed, noted, classified, and evaluated behaviour during each stage of the simulation.

3. Clarified the use of the rating scale in practice.

We now explain in detail how assessors used the six-point scale (A = 0; B = 1; B– = 2; B+ = 3; C– = 4; C = 5) during the VRDC sessions, including:

– scoring principles,

– decision rules for selecting anchors,

– examples of behavioural indicators,

– calibration procedures.

4. Clarified whether all five competencies were assessed during all five stages.

We now clearly explain that each stage elicited all five competencies, but the intensity and behavioural load varied across exercises. The manuscript includes additional clarification on how competencies were mapped to scenario events.

6. Clarified the origin of the competency model.

We now state explicitly that both the VRDC and the 270-degree instrument were based on a competency model developed through a systematic job analysis of the mid-level manager role in the organisation.

2.3 Results

The results of the interrater agreement for both the VRDC and the 270-degree assessments were mentioned in the manuscript, as well as the ICC values. Perhaps tables indicating these results can be included to support the findings by the authors.

Our response:

We thank the reviewer for this valuable suggestion. We have now added two comprehensive tables (Tables 1 and 2) presenting the inter-rater reliability indices for both 270-degree and VRDC assessments. These tables include rwg, ICC(1), and ICC(2) values for all five competencies, along with sample sizes. The tables have been inserted in the "Preliminary analyses" section before the correlation analyses, and we have added corresponding text to interpret these reliability statistics and explain the exclusion criteria applied (rwg < 0.70).

Five tables indicating the results from a bivariate analysis per competency, correlating the VRDC rating; the Self-rating and the Other-rating were included. Each time comments about the interpretation of the results were given. Please add additional comments to the interpretation of each of the five tables. As example, on Table 1 (p11) “Regarding the evaluation of M, employee self-assessments were significantly and negatively correlated with VRDC assessments (r = -0.29, p < 0.05), and others assessments were significantly and positively correlated with VRDC assessments (r = -0.40, p < 0.01).” Add what this means for the individual, as example: the individuals themselves under evaluate their capability on this competency when compared to the VRDC rating, while Others’ rating tends to be more aligned to the VRDC rating. (Note that in the text on p11 the Other-rating r=-0.40, while in Table 1 (p12) it is indicated as 0.40.)

Our response:

We have enhanced the interpretation of all five correlation tables (Tables 3-7) by adding practical explanations of what the correlations mean for individual participants. For example, for Table 3 (Managing People and Tasks), we now explicitly state that participants tend to underestimate their own capabilities compared to VRDC performance, while others' perceptions align more closely with observed behaviors. Similar interpretations have been added for all other competencies, connecting the statistical findings to their practical implications for assessment and development.

We have also corrected the typographical error noted by the reviewer: the text now consistently reports r = 0.40 (not r = -0.40) for the correlation between others' assessments and VRDC in Managing People and Tasks.

Perhaps also specify which bivariate method was used.

Our response:

We have added a clear statement at the beginning of the "Correlation analyses" section specifying that "Pearson product-moment correlation coefficients were calculated to examine the relationships between study variables."

It could perhaps also be useful to add a chart (like a bar-chart) to visually indicate the correlation of the VRDC ratings with the Self and Other ratings per competency. As e

---

## [Editor Report · Decision Letter 1]

14 Dec 2025

Comparison of Virtual Reality Development Centers and 270-Degree Evaluations in the Context of Mid-Level Managers' Competencies

PONE-D-25-08086R1

Dear Dr. Baczynska,

We’re pleased to inform you that your manuscript has been judged scientifically suitable for publication and will be formally accepted for publication once it meets all outstanding technical requirements.

Kind regards,

Saeed Siyal

Academic Editor

PLOS One
---

## [Editor Report · Acceptance letter]

PONE-D-25-08086R1

PLOS One

Dear Dr. Baczyńska,

I'm pleased to inform you that your manuscript has been deemed suitable for publication in PLOS One. Congratulations! Your manuscript is now being handed over to our production team.

Kind regards,

on behalf of

Dr. Saeed Siyal

Academic Editor

PLOS One